# Development of a New Dolomite-Based Adsorbent with Phosphorus and the Adsorption Characteristics of Arsenic (III) in an Aqueous Solution

Zoltuya Khashbaatar [1], Shota Akama [1], Naoki Kano [2] and Hee-Joon Kim [2,*]

1  Graduate School of Science and Technology, Niigata University, Niigata 950-2181, Japan;
   f19k501a@mail.cc.niigata-u.ac.jp (Z.K.); f19b037a@mail.cc.niigata-u.ac.jp (S.A.)
2  Department of Chemistry and Chemical Engineering, Faculty of Engineering, Niigata University,
   8050 Ikarashi 2-Nocho, Nishi-ku, Niigata 950-2181, Japan; kano@eng.niigata-u.ac.jp
*  Correspondence: kim@eng.niigata-u.ac.jp; Tel.: +81-025-262-7538

**Abstract:** In recent decades, the removal of hazardous chemicals that have entered wastewater and groundwater as a result of industrial and consumer activities has become an issue of concern. Specifically, removing arsenic (III) from groundwater is critical and equally crucial in the use of low-cost, efficient adsorbent materials. One purpose of this study was to develop a low-cost hydroxyapatite adsorbent ($Ca_5(PO_4)_3OH$) by reacting the Ca component of calcined dolomite with phosphorus, and another was to apply the developed adsorbent to remove arsenic (III) from well water in developing countries. In this study, phosphorus adsorption was performed on thermally calcined dolomite, and the adsorption isotherm of the phosphorus study was investigated on selected calcined dolomite. The maximum amount of phosphorus on the selected calcined dolomite was 194.03 mg-P/g, and the Langmuir isotherm model was fitted. Arsenic (III) adsorption was investigated in a wide pH range (pH 2~12) using the new adsorbent. The amount of arsenic (III) adsorbed was 4.3 mg/g. The new absorbent could be effective in removing arsenic (III) and become an affordable material.

**Keywords:** dolomite-based adsorbent with phosphorus; calcination; arsenic (III); adsorption isotherm; aqueous solution

## 1. Introduction

In 2015, the United Nations adopted 17 Sustainable Development Goals (SDGs) to address the critical concerns of climate change and environmental degradation, and the efforts to reduce environmental burdens and health hazards are becoming more active worldwide [1]. In terms of harmful substances, such as heavy metals, reducing emissions resulting from production and consumption, reducing water and soil pollution, and purifying groundwater are highly related to Goal 6, Goal 3, and Goal 12 of the SDGs. The development of heavy metal adsorbents for the treatment of well water and wastewater, which is the goal of this research, can contribute to these three goals [1,2].

Recently, in Japan, regulations on heavy metals contained in wastewater and soil have become stricter. Regarding contaminated wastewater, the revision of the Water Pollution Control Law has strengthened the standard values for arsenic (As) and lead (Pb), while cadmium (Cd), selenium (Se), boron (B), and fluorine (F) have been newly designated as regulated substances. New purification treatment methods and secondary treatment methods are being developed to meet these new standards [3,4].

The Japanese government enacted the Water Pollution Law (1970) and the Basic Environment Law (1993), which specified environmental quality standards concerning the contamination of soil and groundwater pollution to preserve public health and the environment. In 2002, the Soil Contamination Countermeasures Act prescribed target chemical substances, investigation, and designated areas for management and deregistration after

completion of remediation to prevent repercussions of contamination of soil and water by, for example, heavy metals. This act classifies hazardous substances into three classes. Class 2 includes nine items (e.g., As, Cr, Se, Pb, F, B, and their compounds), each of which poses a risk on direct ingestion [5].

Furthermore, since the revised Soil Contamination Countermeasures Law in 2010 regulated natural pollution [5], the number of cases designated as countermeasure areas is increasing rapidly. In particular, the proportion of contamination by As, Pb, and F (Class 2 specified hazardous substances) is high, at over 70% [4]. To remove these specified hazardous substances and purify contaminated soil, the excavation treatment method is generally used, but this method is costly. Therefore, attention is being paid to the method of adding a heavy metal adsorbent to the soil to reduce the number of heavy metals eluted from the infiltrated water. Although excavation treatment can prevent the elution of heavy metals, the method has excessive specifications for application to naturally contaminated soil with a low degree of pollution, and is costly. Therefore, there is an increasing need for the adsorbent adding method, which is less costly than excavation treatment.

Arsenic contamination in groundwater threatens over 108 countries, and more than 230 million people have already been treated for arsenic-related issues [6], with the maximum number of people in Bangladesh and West Bengal in India being at dangerous levels of exposure [7]. In natural waterways, arsenic is mainly found in two oxidation states, (III) and (V) [8]. As (V) oxyanions have negative charges and can be removed by an ion-exchange process, whereas As (III) is non-ionic. Prior to its removal, As (III) must be oxidized to As (V) [9]. Iron-based oxide/hydroxides [10] and ozone [8] are used by oxidizing agents, and electrocoagulation is a selective method [11] for removing As (III). Particularly, porous membranes, hybrid membranes processes, electrocoagulation, and adsorption assisted membrane systems have proven to be effective for arsenate adsorption in addition to iron oxyhydroxides adsorption [12–15]. However, due to the additional energy required to filter coagulants rich in As to purify water, it is not practical for use in developing nations [8,9]. Due to the scarcity of connected urban water supply facilities for millions of people [7], adsorption-based remediation using commercially available and low-cost adsorbents is recommended these days [16,17].

Over the last decade, extensive research has investigated removing arsenic using biomasses such as food industry by-products and agricultural wastes as sorbents [18–20]. The removal of arsenic and oxidation of As (III) has been investigated using inexpensive methods that employ bacteria [21] and yeast [22] as oxidizing agents. Therefore, research on biochar-based adsorbents has been conducted; these sorbents are environmentally safe. Nevertheless, due to the lower adsorption amount of arsenic, physical and chemical pretreatment is required [23].

Adsorbents based on carbonate minerals for the removal of arsenic from drinking water have been the target of a lot of research in the development of low-cost and simple removal technologies [24–26]. Some researchers confirm that calcined dolomite, a low-cost natural mineral, shows a high adsorption ability toward boron [27], lead, fluorine [28], and arsenic [28,29]. However, its arsenic absorption ability is low and the re-eluting problem occurs due to a high alkalic medium. The adsorbent developed in this research can handle the above-mentioned weak points and be used to remove heavy metals from groundwater and wastewater. In addition, MgO is one of the main components in arsenic adsorption removal [30], although MgO is soluble in water in neutral environmental conditions (e.g., pH = 7).

Based on Amirthalingam et al. [31], hydroxyapatite with an admixture of Mg ions was obtained with the mechanochemical synthesis of hydroxyapatite by reacting dolomite with diammonium phosphate. Furthermore, the carbonate quantification of apatite minerals was investigated with FT-IR by Grunenwald et al. [32].

The purpose of this study was to develop a dolomite-based adsorbent with phosphorus and to investigate its adsorption characteristics for the removal of arsenic from an aqueous solution. The developed dolomite-based adsorbent with phosphorus is composed of

magnesium oxide (MgO) and hydroxyapatite ($Ca_5(PO_4)_3OH$ or Hap), which has high ion-exchange properties and displays efficient adsorption of arsenic, which makes it an effective adsorbent material for water treatment.

## 2. Materials and Experimental Methods

### 2.1. Dolomite Material

Domestic dolomite was used in this research. Dolomite was sieved (75–150 μm) and dried in a horizontal drying oven at 105 °C for 24 h. Then dolomite samples (approximate mass of 0.5 g) were calcined (decomposed) in the vertical fixed-bed reactor at 300–900 °C temperatures under air, $N_2$, and $CO_2$ input gases. Figure 1 illustrates the schematic diagram of the calcination apparatus in which the decomposition experiments were performed. The temperatures of the electric heater were set in advance, and the mass change of dolomite was continuously measured by a digital balance during the calcination process. The chemical compositions of dolomite and obtained products (calcined dolomites) after calcination were determined by XRD analysis.

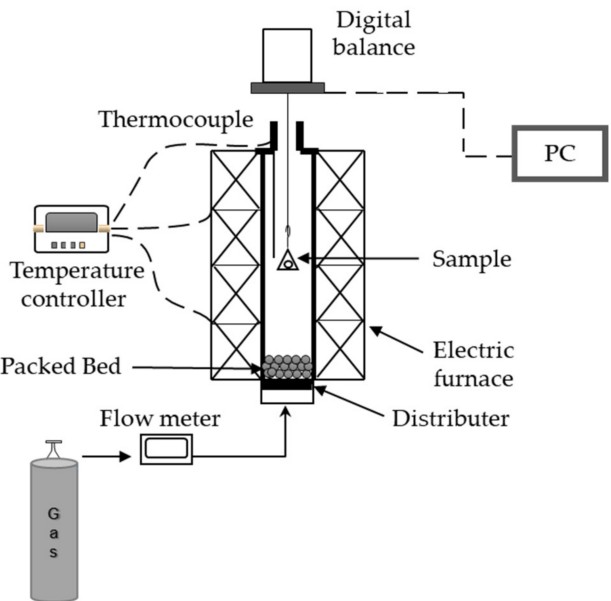

**Figure 1.** Schematic diagram of the fixed-bed reactor for calcination.

### 2.2. Characterization of the Adsorbent

To assess the properties of the re-calcined new adsorbent, the BET surface area was determined by the nitrogen adsorption/desorption measurements, which were employed in TriStarII3020, Micromeritics, and the components of the adsorbent were detected using an X-ray diffractometer (D2 Phaser, Bruker, Billerica, MA, USA).

### 2.3. Adsorption Experiments

#### 2.3.1. Theory of Adsorption

The Langmuir model and the Freundlich model were used to evaluate the mechanism of phosphate adsorption. The isotherm equations are given in Equations (1) and (2):

$$\text{Langmuir} : \frac{1}{q_p} = \frac{1}{q_{max}} + \frac{1}{K_L \times q_{max}} \times \frac{1}{C_e} \tag{1}$$

$$\text{Freundlcih} : \ln q_p = \ln K_F + \frac{1}{n} \ln C_e \tag{2}$$

where $K_L$ is the Langmuir adsorption coefficient (in L/mg), $q_{max}$ is the maximum adsorption amount (in mg-P/g), $K_F$ is the Freundlich constant (in mg-P/g or $(L/mg)^{1/n}$), and $n$ is the Freundlich exponent (–).

2.3.2. Adsorption Experiments of Phosphorus Onto Calcined Dolomite

To characterize the adsorption of phosphorus onto selected calcined dolomite, phosphorus was added to the calcined dolomite as follows: a stock solution of phosphorus with an initial concentration of 2000 ppm was prepared by dissolving accurate amounts of $H_3PO_4$ in deionized water. This stock solution was used to create more solutions of phosphorus by dilution. The pH of the solution was adjusted by using 5M NaOH. The condition of isotherm experiments was 2.5 g/L dosage of calcined dolomite and different initial concentrations of phosphorus. The mixture was stirred at room temperature for 24 h. Then, solid and the liquid was separated by a vacuum aspirator. Finally, the concentrations of phosphate in the supernatant solution after and before adsorption were determined by a molybdenum blue spectrophotometric method at the wavelength $\lambda = 880$ nm. The adsorption amount of phosphorus (mg/g) was calculated using the following equation:

$$q_p = (C_0 - C_e)V/m \tag{3}$$

where $q_p$ is the adsorption amount (in mg-P/g), $C_0$ and $C_e$ are the initial and equilibrium concentrations of phosphorus (in mg/L), $V$ is the volume of the solution (in L), and $m$ is the mass of calcined dolomite (in g).

2.3.3. Adsorption Experiment of As (III) Onto a Re-Calcined Adsorbent with Phosphorus

Adsorption experiments were conducted to evaluate the impact of using a re-calcined adsorbent with phosphorus for As (III) removal. First, a 1000 ppm stock solution of As (III) was prepared by dissolving an adequate amount of $As_2O_3$ and diluting it with deionized water. Adsorption of As (III) in different mediums was investigated using a 2 g/L adsorbent dose in the initial solution of 10 ppm for 24 h at room temperature. After achieving equilibrium, samples were filtered by vacuum aspirator using a filter (GF/B, GE Healthcare Life Sciences Co., Ltd., Chicago, IL, USA) The filtrate was acidified with 3% $HNO_3$, and the arsenic concentration in the filtrate was measured by ICP–MS (Thermo Fisher Scientific. X–SERIES II, Waltham, MA, USA). The adsorption of As (III) amount was calculated by Equation (4).

$$q_{As} = (C_0 - C_e)V/m \tag{4}$$

where $q_{As}$ is the adsorption amount (in mg-As/g), $C_0$ and $C_e$ are the initial and equilibrium concentrations of As (III) (in mg/L), $V$ is the volume of the solution included As (in L), and $m$ is the mass of adsorbent (in g).

## 3. Results and Discussion

### 3.1. Characterization of the Calcination Process

This study employed raw dolomite (from Sano City, Japan) with a particle size of 75–150 μm and a specific surface area of 0.4171 $m^2$/g. Table 1 shows the main chemical components of dolomite.

**Table 1.** Chemical analysis results of dolomite (wt%).

| Ignition loss | CaO | MgO |
|---|---|---|
| 46.76 | 34.11 | 17.92 |

The kinetic curves of calcination in different atmospheres of raw dolomite are shown in Figure 2. The weight loss was about 47% in air and $N_2$ input gases at 800 °C, which confirms that dolomite completely calcined into CaO and MgO. In contrast, the weight

loss in the $CO_2$ atmosphere was approximately 20% due to the calcination of only $MgCO_3$, which leads to the first step or half-decomposition of dolomite [33].

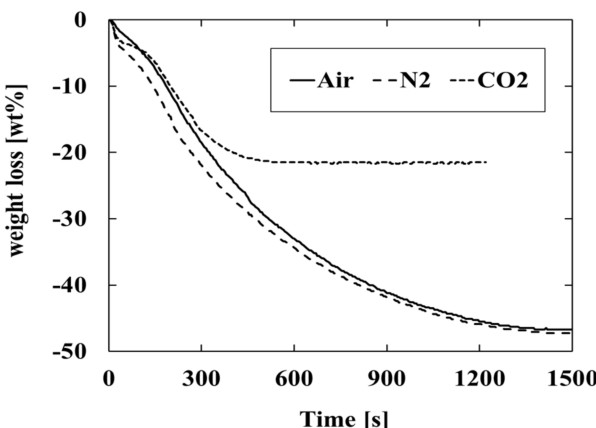

**Figure 2.** Kinetic curves of dolomite at 800 °C in different input gases.

X-ray diffraction patterns of raw dolomite and calcined dolomites are illustrated in Figure 3. XRD results reveal that raw dolomite consisted of dolomite ($CaMg(CO_3)_2$) and calcite ($CaCO_3$). The dolomite peak disappeared and was replaced by peaks of CaO and MgO at calcination under 800 °C in an air atmosphere. However, $CaCO_3$ and MgO peaks and additional $CaCO_3$ peaks appeared in the $CO_2$ atmosphere. The explanation is that $CO_2$ is absorbed by the generated CaO in calcined dolomite under the $CO_2$ environment, resulting in $CaCO_3$ [34] and the extra calcite peaks that appear in the XRD pattern. Former researchers have reported that dolomite is calcined in two stages. First, $CaCO_3$, MgO, and $CO_2$ are generated (Equation (5)). Then, $CaCO_3$ is decomposed into CaO and $CO_2$ (Equation (6)). However, phase composition and morphology are significantly influenced by the calcination temperature and the input gases [34–36].

$$CaMg(CO_3)_2 \rightarrow CaCO_3 + MgO + CO_2 \tag{5}$$

$$CaCO_3 \leftrightarrow CaO + CO_2 \tag{6}$$

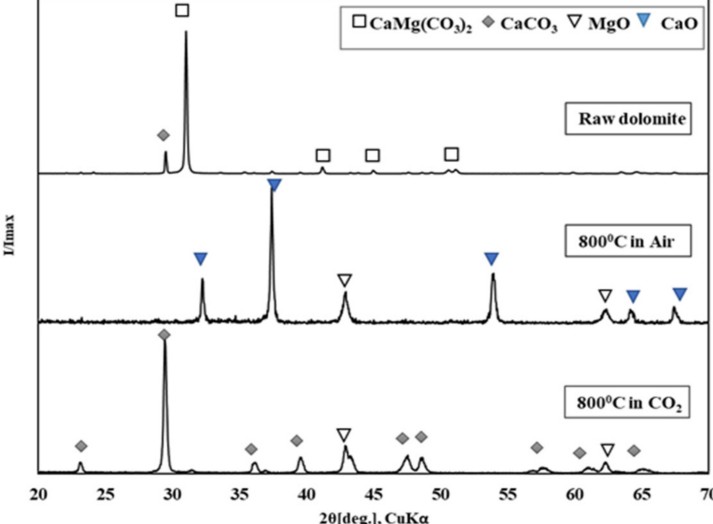

**Figure 3.** XRD results of raw dolomite and calcined dolomites.

According to the reports of Olszak–H et al. [36], the calcination of dolomite in Equations (5) and (6) occurs above 588.5 and 1118.8 K under an air input gas, respectively.

### 3.2. Characterization of Calcined Dolomite with a Phosphorus Component

#### 3.2.1. Adsorption of Phosphorus Onto Calcined Dolomite

The adsorption amount of phosphorus onto different calcined dolomites was investigated at various contact times. The addition of phosphorus to calcined dolomite in the alkalic medium can be shown by Equation (7) [37].

$$Ca^{2+} + PO_4^{3-} + OH^- \rightarrow Ca_5(PO_4)_3OH \tag{7}$$

Figure 4 shows the kinetic curve of adsorption of phosphorus onto selected calcined dolomite. The adsorption equilibrium was reached after 6 h. However, subsequent experiments were carried out for 24 h to give sufficient equilibrium time.

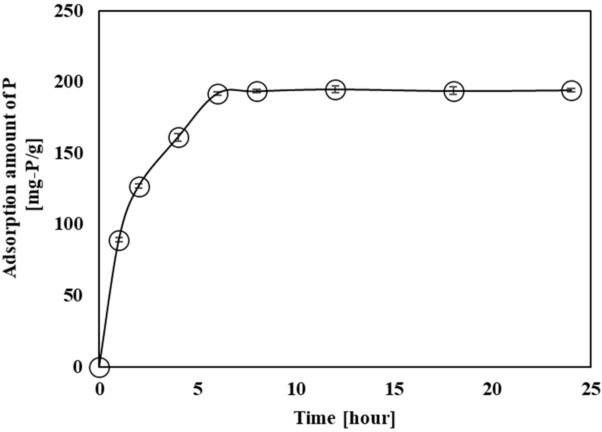

**Figure 4.** The kinetic curve of the adsorption of phosphorus onto calcined dolomite.

#### 3.2.2. Adsorption Isotherm

The Langmuir and Freundlich linearized models [38] were used to evaluate the experimental data, and the outcomes are provided in Table 2 and Figure 5, respectively. To ensure the accuracy of the data-fitted models, the mean relative errors (MREs) were calculated [39].

**Table 2.** Isotherm parameters of Langmuir and Freundlich for the adsorption of phosphorus onto calcined dolomite.

| Langmuir Model | | | | Freundlich Model | | | |
|---|---|---|---|---|---|---|---|
| $q_{max}$ (mg/g) | $K_L$ (L/mg) | $R^2$ | MRE (%) | $K_F$ (mg/g) | $\frac{1}{n}$ | $R^2$ | MRE (%) |
| 194.03 | 1.291 | 0.9837 | 0.5677 | 83.39 | 0.1548 | 0.6732 | 5.70 |

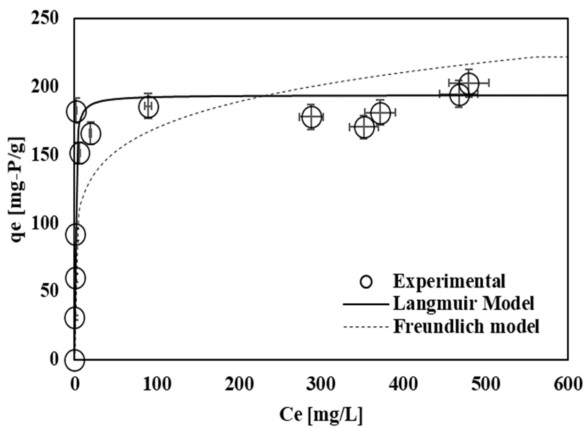

**Figure 5.** Adsorption isotherm of phosphorus on calcined dolomite.

With a correlation coefficient $R^2$ of 0.9837, the Langmuir isotherm fitted to the experimental data, as seen in Table 2 and Figure 5. The adsorbate adsorbs as a single layer on the surface of the adsorbent according to the Langmuir isotherm model [30]. As a result, we assumed a prominent monolayer of phosphorus adsorbed on the surface of calcined dolomite. The maximum adsorption amount of phosphorus onto calcined dolomite was 194.03 mg-P/g by the Langmuir fitting model.

### 3.2.3. Characterization of a Dolomite-Based Adsorbent with Phosphorus

We re-calcined the dolomite-based adsorbent with phosphorus for 30 min at 340–800 °C under an air input gas to promote the porosity of the adsorbent. Table 3 shows BET surface area, pore volume, and pore size of the different re-calcined adsorbents. Thermal modification reveals that the surface area, the pore volume, and the pore size increased as the temperature increased from 100 °C to 360 °C. However, these parameters decreased at 800 °C. According to the viewpoint of Stefaniak et al. [40], the high temperature leads to the increased density of the solid, so the pore volume and the specific surface area decrease due to shrinkage and the sintering process.

**Table 3.** Characterization of the dolomite-based adsorbent with phosphorus in different re-calcination temperatures.

| Re-Calcination Temperature (°C) | BET Surface Area (m²/g) | Pore Volume (cm³/g) | Pore Size (nm) |
| --- | --- | --- | --- |
| 100 | 64.3155 | 0.1597 | 9.9365 |
| 360 | 67.1562 | 0.1885 | 11.2326 |
| 800 | 23.0791 | 0.0545 | 9.4565 |

On the basis of the findings of the re-calcination experiments, in subsequent As (III) adsorption experiments, we employed a dolomite-based adsorbent with phosphorus re-calcined at 360 °C under an air input gas.

### 3.2.4. Characteristics of As (III) Adsorbed Onto a Dolomite-Based Adsorbent with Phosphorus

Adsorption experiments of As (III) were varied at different pH levels using a dolomite-based adsorbent with phosphorus re-calcined at 360 °C, and the results are shown in Figure 6.

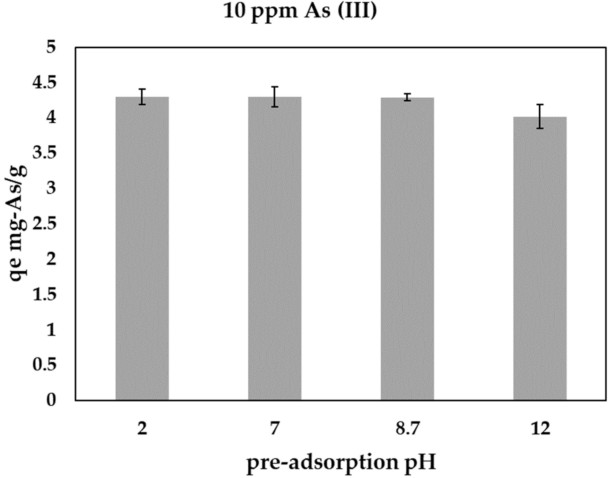

**Figure 6.** Adsorption capacity of As (III) onto a dolomite-based adsorbent with phosphorus for varying pre-adsorption pH.

The adsorption of arsenic species strongly depends on pH because As (III) commonly dissolves as neutrally charged in the form of $H_3AsO_3$ at pH < 9 and As (III) dissociates into

its anions $H_2AsO_3^-$, $HAsO_3^{2-}$, and $AsO_3^{3-}$ at pH > 9 [25]. Thus, we assume that the As (III) adsorption amount decreases in an alkaline condition of pre-adsorption pH > 8.7 due to the formation of negatively charged ions of arsenic. Therefore, the repulsion force between arsenite and phosphate anions can be the reason for a decrease in the adsorption amount of arsenic in high pre-adsorption pH. Table 4 compares the results of As (III) adsorption on various adsorbents, including the new adsorbent developed in this study.

**Table 4.** Comparison of characteristics of As (III) adsorbed onto different adsorbents.

| Adsorbent | Solid/Liquid Ratio (g/L) | pH | Initial Concentration (mg/L) | Adsorption Amount (mg/g) | References |
|---|---|---|---|---|---|
| Bone Char | 5 | 8.6 | 0.05–50 | 2.659 (Sips) | [41] |
| Lignite | 10 | 2.5–2.7 | 1–50 | 0.324 (Langmuir) | [42] |
| Bentonite | 10 | 6.2–6.7 | 1–50 | 0.317 (Langmuir) | [42] |
| Rice Husk | 1 | 7 | 0.1–1 | 0.775 (Langmuir) | [43] |
| Coconut Shell | 20 | 10 | 0.5 | 0.3688 (Real) | [44] |
| Charred Dolomite | 1 | 7.2 | 50–200 | 1.846 (Langmuir) | [26] |
| A dolomite-Based Adsorbent with P | 2 | 2 | 10 | 4.301 (Real) | This study |
| A dolomite-Based Adsorbent with P | 2 | 8.7 | 10 | 4.295 (Real) | This study |

Bone char mainly consists of hydroxyapatite [45], so we can assume that the dolomite-based adsorbent with phosphorus used in this study has the same adsorption mechanism. As shown in Table 4, the As (III) adsorption amount was higher (4.29 mg-As/g) for the dolomite-based adsorbent with phosphorus used in this study than that for bone char (2.659 mg-As/g) at pre-adsorption pH = 8.7, though the initial As (III) concentrations were different for the adsorbents. However, As (III) adsorption cannot be attributed only to the hydroxyapatite component, and another important component can be MgO. Some researchers [28] reported that the local structure around Ca atoms did not change during calcination but that the local structure of Mg atoms changed gradually and the porosity of MgO can be improved by increasing the adsorption amount of As (III). The new adsorbent we have developed is mainly composed of $Ca_5(PO_4)_3OH$ and MgO, so it can be good at ion exchange and is a high porosity material, which can be good characteristics for removing heavy metals.

*3.3. Conclusions*

We developed a dolomite-based adsorbent with phosphorus for heavy metal removal that could contribute to SDG Targets 3, 6, and 12, and investigated their applicability as an arsenic removal adsorbent. We drew the following conclusions:

1. The composition of calcined dolomite is controlled by the calcination reaction time and the input gas;
2. Phosphorus adsorption onto calcined dolomite was fitted with the Langmuir method, and the precipitated phosphorus interacts with CaO parts, forming $Ca_5(PO_4)_3OH$;
3. A dolomite-based adsorbent with phosphorus is an effective adsorbent material for arsenic removal;
4. Further studies should investigate arsenic adsorption experiments in groundwater and wastewater and reveal the mechanism of arsenic removal.

**Author Contributions:** Experiment, data curation, instrument measurements, and writing, Z.K. and S.A.; writing—draft preparation H.-J.K. and Z.K.; writing—review and editing, H.-J.K. and N.K.; supervision H.-J.K. and N.K. All authors have read and agreed to the published version of the manuscript.

**Funding:** This work was supported by the Uchida Energy Science Promotion Foundation and the Sasaki Environmental Technology Foundation.

**Institutional Review Board Statement:** Not applicable.

**Informed Consent Statement:** Not applicable.

**Data Availability Statement:** Not applicable.

**Conflicts of Interest:** The authors declare no conflict of interest.

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
