# Peer review of "Development of a New Dolomite-Based Adsorbent with Phosphorus and the Adsorption Characteristics of Arsenic (III) in an Aqueous Solution"

_water, doi:10.3390/w14071102_

Round 1
Reviewer 1 Report
The work presented to me for review, prepared by Z. Khashbaatar, S. Akam, N. Kano and H.-J. Kim entitled "Development of a New Dolomite-Based Adsorbent with Phosphorus and the Adsorption Characteristics of Arsenic (III)" presents a new method of obtaining the adsorbent of As (III) compounds from dolomite. The process of preparing the adsorbent in the first stage consists in calcination of dolomite in order to obtain calcium and magnesium oxides, and then adsorption of phosphate ions from the phosphoric acid solution. The obtained product was dried and then calcined for 30 minutes in an air atmosphere at a temperature of 300-800 ° C. Calcined dolomite and end product samples were subjected to XRD structural analysis as well as porosity and specific surface analysis. Based on this analysis, the authors concluded that the obtained product has the structure of hydroxyapatite. The kinetics of adsorption of As (III) compounds and adsorption as a function of solution pH and As (III) concentration were investigated by the method of As (II) concentration loss in solution, As (III) concentration was determined by the ICP_MS method.
After reading the manuscript, I have the following comments and doubts:
Page 1 line is "Arsenic (III) adsorption was investigated in a wide pH range (pH 2 ~ 12) using the new adsorbent". Hydroxyapatite is a partially soluble compound, its solubility significantly changes with the pH of the solution. Does the mentioned measuring range refer to the pre-adsorption pH or the equilibrium adsorption pH? Did equation 4 take into account the change in adsorbent mass due to solubility as a function of pH in the adsorption calculations?
Hydroxyapatite is a compound that can substitute both Ca cations and PO43- or OH- anions in the structure. Due to the fact that during the formation of hydroxyapatite, Mg2 + ions were present in the system, which can replace Ca ions in position II of the hydroxyapatite crystal structure, it would be necessary to analyze the diffractograms of the obtained hydroxyapatite samples using Rietweld's method in order to determine the content of HAp, MgO, CaO. As a result of the mechanochemical synthesis of hydroxyapatite by reacting dolomite with diammonium phosphate N. Amirthalingam et al. Materials Letters 254 (2019) 379–382 received hydroxyapatite with an admixture of Mg ions. I believe that the authors of the manuscript should compare the results of their synthesis with the results presented in the work of N. Amirthalingam et al.
Another problem concerning the characteristics of hydroxyapatite is related to the presence of CO2 in the air during syntheses and calcination, the TG curve in Fig. 2 shows that the weight loss of calcined dolomite in air is less than that of the sample calcined in nitrogen atmosphere, which proves that there is some carbonate residue. FTIR measurements should be made to determine the carbonate content of HAp, see Grunenwald, A .; Keyser, C .; Sautereau, A.M .; Crubézy, E .; Ludes, B .; Drouet, C. Revisiting carbonate quantification in apatite (bio) minerals: A validated FTIR methodology. J. Archaeol. Sci. 2014, 49, 134–141.
References should be prepared in accordance with the Editorial Board's recommendations.
After the corrections, I believe that the work can be published in the Water journal.
Author Response
To Reviewer 1
Thank for your comments and suggestions. We write reply for your comments. Please see the followings;
Comments:
(1) Page 1 line is "Arsenic (III) adsorption was investigated in a wide pH range (pH 2 ~12) using the new adsorbent". Hydroxyapatite is a partially soluble compound, its solubility significantly changes with the pH of the solution. Does the mentioned measuring range refer to the pre-adsorption pH or the equilibrium adsorption pH? Did equation 4 take into account the change in adsorbent mass due to solubility as a function of pH in the adsorption calculations?
Response: Thank you for your comment and valuable suggestion. In this manuscript, the pH means pre-adsorption pH. The mass change before and after adsorption was small in this experiment, so we calculated the adsorption amount by using equation 4 in this study. As your comment, at further paper, we will measure the mass change more precisely and take in account the change in adsorbent mass due to solubility as a function of pH for calculating the adsorption amount.
(2) Hydroxyapatite is a compound that can substitute both Ca cations and PO43- or OH- anions in the structure. Due to the fact that during the formation of hydroxyapatite, Mg2 + ions were present in the system, which can replace Ca ions in position II of the hydroxyapatite crystal structure, it would be necessary to analyze the diffractograms of the obtained hydroxyapatite samples using Rietweld's method in order to determine the content of Hap, MgO, CaO. As a result of the mechanochemical synthesis of hydroxyapatite by reacting dolomite with diammonium phosphate N. Amirthalingam et al. Materials Letters 254 (2019) 379–382 received hydroxyapatite with an admixture of Mg ions. I believe that the authors of the manuscript should compare the results of their synthesis with the results presented in the work of N. Amirthalingam et al.
Response: Thank you for your comment. We also think that your suggestion is very important and interesting matter. Then the description about mechanochemical synthesis conducted by N. Amirthalingam was added in “Introduction” as references [31] in the modified manuscript. However, this manuscript put most important point for the adsorption of arsenic by the novel adsorbent. Then the detail mechanism or the comparison of hydroxyapatite content among synthesis methods will be presented elsewhere in the future paper.
(3) Another problem concerning the characteristics of hydroxyapatite is related to the presence of CO2 in the air during syntheses and calcination, the TG curve in Fig. 2 shows that the weight loss of calcined dolomite in air is less than that of the sample calcined in nitrogen atmosphere, which proves that there is some carbonate residue. FTIR measurements should be made to determine the carbonate content of HAp, see Grunenwald, A .; Keyser, C .; Sautereau, A.M .; Crubézy, E .; Ludes, B .; Drouet, C. Revisiting carbonate quantification in apatite (bio) minerals: A validated FTIR methodology. J. Archaeol. Sci. 2014, 49, 134–141.
Response: Thank you for your comment. We also think that your suggestion is very important and interesting matter. Then the description about FTIR measurements by Grunenwald, A. et al. was slightly added in “Introduction” as references [32] in the modified manuscript. On the other hand, after 1200s progress, the weight loss of calcined dolomite between in air and in nitrogen atmosphere is not so large. Moreover, the adsorption of arsenic by the novel adsorbent was mostly put weight in this manuscript, and we want to reduce the description about phosphoric adsorption. Then detail characteristics of hydroxyapatite including the amount of carbonate residue will be presented elsewhere in the future paper.
(4) References should be prepared in accordance with the Editorial Board's recommendations.
Response: Thank you for your comment. Based on your comment, we have revised the references.
(5) After the corrections, I believe that the work can be published in the Water journal.
Response: Thank you for your comments. The comments are highly encouraging and helpful for our manuscript improvement. We are aware of the fact that there are still many shortcomings in our manuscript. We have checked the manuscript again and modified some parts which you pointed out.

Reviewer 2 Report
Please see my comments in the word file.

Author Response
To Reviewer 2
Thank for your comments and suggestions. We write reply for your comments. Please see the followings;
General comment:
The manuscript is devoted to removal of arsenite ion from aqueous solution using low-cost adsorbent as a hydroxyapatite. The material characterization is partially performed. The adsorption performance of the material for phosphorus is determined using batch adsorption tests and kinetic and equilibrium data is obtained. Their findings are interesting, and original, overall, the manuscript is relatively well-written and brings valuable information. However, my major concern is about investigations on the phosphorus adsorption using the adsorption. The introduction is focused on arsenite pollution but the most of the presented results are on phosphorus adsorption. Second concern is about the important parameter of the adsorbent, which is the surface charge of the adsorbent and is not determined in this work. In addition, there are some suggestions for the revision of the manuscript.
Response: Thank you for your comments. The comments are highly encouraging and helpful for our manuscript improvement. We are aware of the fact that there are still many shortcomings in our manuscript. We have checked the manuscript again and modified or left out some parts which you pointed out.
Specific Comments:
- Keywords are not suitable for the presented work.
Response: Thank you for your comment. Based on your comment, we have revised the keywords.
- Lines 31-38: the source of the information is lacking.
Response: Thank you for your comment. Based on your comment, we have added the source of the information regarding SDGs as references [1] and [2] in the modified manuscript.
- Line 59-65: If the adsorbent added to soil is fully saturated, how the adsorbent can be taken out/separated from the soil or regenerated? Moreover, it would cause the secondary pollution of the groundwater by leaching of toxic pollutants from saturated adsorbent. So the authors claim of adding adsorbent to soil to adsorbent arsenite is not realistic approach.
Response: Thank you for your comment. We have understood and agreed with your idea. However, in our present research, we don’t think regenerating the adsorbents and separating adsorbents from the soil. It is important for arsenic to be insolubilized and not to leach from adsorbents after arsenic had once adsorbed on the dolomite-based adsorbent.
- Lines 72-73: Porous membranes, hybrid membranes processes, electrocoagulation, adsorption assisted membrane systems (see for instance “Journal of hazardous materials 400(2020), 123221”, “Journal of Chemical Technology & Biotechnology, 96(6) (2022), 1504-1514”, “Chemie Ingenieur Technik, 93(9) (2021), 1396-1400” and “Water 12(2020), 2876” have proven to be effective for arsenate adsorption in addition to iron oxyhydroxides adsorption.
Response: Thank you for your comment. Based on your comment, we have added the removal method of your suggestion in “Introduction” as references [12]-[15] in the modified manuscript.
- Line 93: MgO is soluble in water in neutral environmental conditions (e.g., pH 7).
Response: Thank you for your comment. We also agree your idea. Then we have added the following description “although MgO is only little soluble in water in neutral environmental conditions (e.g., pH 7)” in this revised manuscript.
- Figure 6: What is shown in this figure? The adsorption isotherms of phosphorous are already shown in figure 5.
Response: Thank you for your comment. As you pointed out, Fig. 6 only showed a part of the result of Fig. 5 more clearly (i.e., not bring a new information) to derive some parameters shown in Table 2. Then we have deleted Fig. 6 in this revised manuscript.
- IN the manuscript, the phosphorus adsorption is discussed. At pH 8.7, phosphorus is predominately exist as orthophosphate. I would suggest the authors to clearly write the exact phosphorus speciation in the whole manuscript.
Response: Thank you for your comment. Phosphorus coexist as H3PO4 and H2PO4- below pH 5 (e.g., pH 2), coexist as H2PO4- and HPO42- between pH 5-10 (e.g., pH 7), and coexist as HPO42- and PO43- above pH 10 (e.g., pH 12). At pH 8.7, HPO42- is more dominant. However, this manuscript put most important point for the adsorption of arsenic by the novel adsorbent, and we want to reduce the description about phosphoric adsorption. Then the description about phosphorus speciation was not stated in this manuscript, and detail description of phosphoric adsorption including phosphorus speciation will be presented elsewhere in the future paper.
- Table 4: The adsorption capacities of the studied adsorbent for As(III) taken from literature are, I reckon, Langmuir adsorption capacities calculated from adsorption isotherm data and you have compared the adsorption capacity at single liquid to solid ratio and therefore the comparison is not true.
Response: Thank you for your comment. We also consider that the comparison in Table 4 is not necessarily true because the values may vary liquid to solid ratio or initial concentration. Moreover, the values differ if real value based on experiment or the value calculated based on Langmuir or Sips isotherm value, although it becomes some indication. Then we have retouched and revised Table 4.

Round 2
Reviewer 1 Report
After reviewing the revised version of the manuscript prepared by: Z. Khashbaatar, S. Akam, N. Kano and H.-J. Kim entitled "Development of a New Dolomite-Based Adsorbent with Phosphorus and the Adsorption Characteristics of Arsenic (III)" and the Authors' responses to my comments, I say that some of my comments were taken into account.
The authors in reply to the question: "Does the mentioned measuring range refer to the pre-adsorption pH or the equilibrium adsorption pH?" They stated that, "In this manuscript, the pH means pre-adsorption pH." Therefore, in the manuscript in section 3.4 and in Figure 6, use pH pre-adsorption in place of "pH".
My remarks on the substitution of Mg2 + ions for Ca2 + ions in the hydroxyapatite lattice and the presence of carbonates in the HAp lattice aimed at precise characterization of the adsorbent sample. In the revised manuscript, the Authors cited the work, I mentioned in the review, but did not attempt to better characterize the HAp sample, stating that they would do so in a future work.
I believe that after the correction of Fig. 6 and the discussion that it is about pH before adsorption, the work can be accepted for publication in the journal Water.
Author Response
Comments:
(1) The authors in reply to the question: "Does the mentioned measuring range refer to the pre-adsorption pH or the equilibrium adsorption pH?" They stated that, "In this manuscript, the pH means pre-adsorption pH." Therefore, in the manuscript in section 3.4 and in Figure 6, use pH pre-adsorption in place of "pH".
Response: Thank you for your valuable suggestion. Based on your comment, we have revised from "pH" to “pH pre-adsorption” in Section 3.4 and in Figure 6 in re-revised manuscript.
(2) My remarks on the substitution of Mg2+ ions for Ca2+ ions in the hydroxyapatite lattice and the presence of carbonates in the HAp lattice aimed at precise characterization of the adsorbent sample. In the revised manuscript, the Authors cited the work, I mentioned in the review, but did not attempt to better characterize the HAp sample, stating that they would do so in a future work.
Response: Thank you for your comment. In the future work, we are going to conduct more precisely characterization of the sample based on your valuable suggestion.
(3) I believe that after the correction of Fig. 6 and the discussion that it is about pH before adsorption, the work can be accepted for publication in the journal Water.
Response: Thank you for your comments. The comments are highly encouraging and helpful for our manuscript improvement. We have re-checked the manuscript again and modified some parts (in blue letters) which you pointed out.

Reviewer 2 Report
The revised manuscript has taken into account the suggestions of the reviewers and can be accepted.
Author Response
Comment:
The revised manuscript has taken into account the suggestions of the reviewers and can be accepted.
Response: Thank you for your comments. The comments are highly encouraging.
